# A Study on the Development of a White Light Source Module for a Large-Capacity Searchlight Using a Blue Laser Diode

Seung-Min Lee [1] , Joon-Ho Cho [2] and Wan-Bum Lee [3,*]

1   Goodi-Tech Co., Ltd., 26, Yongyeon-gil, Iksan-si 54531, JeonBuk, Republic of Korea
2   Department of Electronics Convergence Engineering, Wonkwang University,
    460 Iksandae-ro, Iksan-si 54538, JeonBuk, Republic of Korea
3   Department of Computer Software Engineering, Wonkwang University,
    460 Iksandae-ro, Iksan-si 54538, JeonBuk, Republic of Korea
*   Correspondence: lwbwon@wku.ac.kr

**Abstract:** In this paper, a large-capacity white light source module using a high-power blue laser diode and a reflective spaced phosphor was designed. The reflective spacing phosphor ensured thermal stability. The proposed white light module is a reflective phosphor structure with a bi-directional optical system based on a rhombus prism lens. The rhombus prism optical system can greatly narrow the blue laser beam width and a long-wavelength band-pass filter of 500 nm or more is applied to change the movement path of the laser beam and transmit white light excited by the phosphor. A dichroic filter was applied to the fold mirror part and a planar convex lens was designed to focus the blue laser beam so that the phosphor was irradiated. Finally, a high-power white light source is obtained from the pre-optics unit to which the dichroic filter is applied. In order to use the proposed white light source as a searchlight, a divergence angle of 4° or less is required. For this implementation, a large-area collimating lens combining an aspheric condensing lens and an achromatic lens was applied. It was confirmed that the divergence angle of 4 degrees or less was satisfied at the focal length (FL) of 38.5 to 42.5 mm of the optical lens of the laser white light module emitter and the collimating optical system.

**Keywords:** blue laser diode; remote phosphor; white module; optical design

## 1. Introduction

Artificial light sources have been developed in the order of incandescent and halogen bulbs that emit light by heat, HID (high-intensity discharge) lamps that emit light by exciting gas, LED (light-emitting diode), and LD (laser diode) applied with semiconductor lighting technology.

In particular, the development of semiconductor lighting technology has made it possible to develop high-performance and functional products depending on the applied technology, and user-centered interface and design have become important factors to consider in product development. Recently, the development direction of lighting is being developed according to the requirements of efficiency, environment, safety, convenience, and design, along with new technological development.

Among semiconductor lighting technologies, LED is applied to various lighting fields owing to its high efficiency, optical safety, and reliability [1–3]. White LED is a solid-state light source with environmentally friendly characteristics and has excellent stability and low power consumption compared with light sources such as conventional incandescent lamps and fluorescent lamps. In addition, it is emerging as a next-generation solid-state lighting because energy consumption efficiency can be drastically improved through advantages such as a long lifespan of more than 100,000 h. White LED lighting has the advantage of exhibiting high light efficiency at a low applied current, but has a disadvantage in that the efficiency decreases as the current increases [4,5]. Many studies are being conducted to

improve these drawbacks; one of them is GaN-based LD lighting. LD has a characteristic in which the external quantum efficiency (EQE) increases linearly as the applied current increases [6]. In addition, SCP (single crystal phosphor) can be excited to obtain high luminous flux and, compared with LED, the mounting area of the chip can be significantly reduced, making it possible to miniaturize and provide sufficient brightness with maximum efficiency [7].

A method of realizing white light using R/G/B laser diodes is being applied and studied in the fields of displays, projectors, and visible light communication. However, when white light is implemented with a short wavelength light source, it is difficult to implement the reproducibility of the wavelength of the white light source [8–10].

A white light source using LD can be implemented as a phosphor plate, and the light efficiency characteristics vary greatly depending on the phosphor material, particle size, and composition ratio [11–15]. In addition, the thermal saturation of the phosphor due to the high power of LD affects the concentration, quantum efficiency, thickness, and so on of the phosphor and eventually damages the phosphor [16–18].

The light characteristics are to satisfy the color of light, the consistency of color, and the ability to maintain the luminous flux over time. This characteristic can be determined by light emission according to the thickness of the phosphor and the angle of laser incidence [9,18,19]. As a way to improve the thermal stability and light characteristics of phosphors, various studies are being conducted, such as changing the transmission type and reflection type structure [20,21]. In addition, research is being conducted on optical structure design methods of white light modules using lasers to ensure efficient light efficiency and stability of optical components [22–25].

A 700 W class LED spot light system recently developed for long-distance irradiation has reached a 10 km luminous distance with a divergence angle of 1.6° [26]. This was possible with high efficiency, high performance, and stable optical components and light distribution design. In addition, the simplification of optical components can improve light efficiency deterioration owing to the use of complex components, and alignment, size reduction, and clear beam patterns of optical components have become possible.

In this paper, a white light module with a new structure was developed using a high-power blue laser and a remote phosphor, and based on an optical structure design using a rhomboid prism lens and a dichroic filter.

In addition, we would like to propose a narrow-angle secondary optical lens design for long-distance propagation, research on key elements of a laser searchlight with thermal and optical stability, and system design.

This paper is organized as follows: Section 1—Introduction, Section 2—Light Source Design (Section 2.1—White Light Module Optical Design, Section 2.2—Searchlight Collimated Beam Optical Design) and Section 3—Conclusions.

## 2. Light Source Design

### 2.1. White Light Module Optical Design

The design of the white light source module using a blue laser diode was simulated using LightTools, and the basic structure consists of a light source part, a transverse displacement part, a fold mirror part, and pre-optics, as shown in Figure 1. The light source consists of three collimated blue LDs. The transverse displacement part was composed of two rhomboid prisms and the fold mirror was composed of a dichroic filter. Figure 1a shows the beam propagation path of the blue laser diode; the blue laser beam diverging on both sides changes its propagation path through a rhombus prism and is irradiated to the dichroic filter, and the blue laser beam located in the center is irradiated directly to the dichroic filter. The beam irradiated to the dichroic filter changes its propagation path by reflection and is focused through the pre-optics. It is a structure that is irradiated to the diffusion part after passing through.

Figure 1b shows the light propagation path of the phosphor excitation and diffusion parts. The light excited by the blue laser beam is emitted as light in a restricted form by the

pre-optics, passes through the dichroic filter, and diffuses parts. The diffused blue laser beam is emitted as light in a limited form by pre-optics, reflected by the dichroic filter, and emitted to the outside of the white light source module to realize white light.

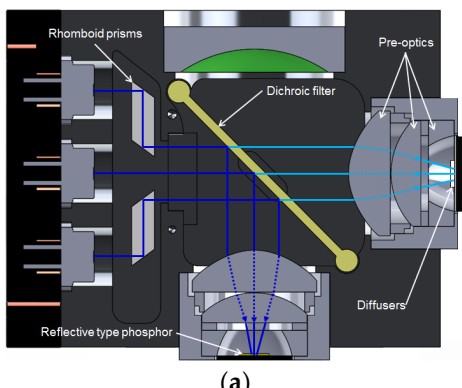

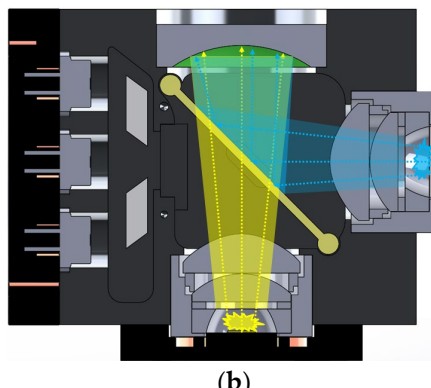

(**a**)  (**b**)

**Figure 1.** Basic optical structure of the white light source module. (**a**) Blue laser diode beam propagation path; (**b**) light propagation path of phosphor excitation and diffusion components.

### 2.1.1. Blue LD Beam Profile Measurement

For the laser diode used to implement the white light source, Nichia's 12 W class blue LD with high power multi-mode characteristics was selected. The specifications of the selected blue LD are shown in Table 1. When an experimental temperature of 25 °C and a forward current of 3.0 A were applied, an optical output of about 4.35 W and a wavelength characteristic of about 455 nm were exhibited. The horizontal optical axis of the divergence angle characteristic is about 0.85°, and the vertical optical axis represents a half-power angle of about 0.1°.

**Table 1.** Blue LD electrical and optical characteristics (Tc = 25 °C, $I_f$ = 3.0 A, CW operation).

| Item | | Symbol | Min. | Typ. | Max. | Unit |
|---|---|---|---|---|---|---|
| Optical Output Power | | Po | - | (4.35) | - | W |
| Dominant Wavelength | | λd | 448 | (455) | 462 | nm |
| Threshold Current | | Ith | 280 | - | 480 | mA |
| Slope Efficiency | | η | - | (1.7) | - | W/A |
| Operating Voltage | | Vop | 3.6 | - | 4.8 | V |
| Beam | Parallel | $\theta_{//}$ | 0.65 | (0.85) | 1.05 | ° |
| Divergence | perpendicular | $\theta_{\perp}$ | −1.0 | (0.1) | 1.0 | ° |

Figure 2 shows the beam profile result of a single blue laser diode, which was measured using a CMOS-1202 product from CINOGY Technologies. For driving the blue laser diode, a CW (continuous wave) type forward current of 3.0 A was applied, the distance between the light source and the sensor was 140 mm, and the ND filter was installed at the 80 mm point between the light source and the sensor. The beam pattern of the blue laser diode was confirmed to have a Gaussian distribution and, as a multi-mode characteristic, confirmed that many mode numbers appeared in the Figure 2a X cross section. The optical power of the blue laser diode was confirmed to be 4.14 W (junction temperature 25 °C) and the optical power was confirmed to be 4.00 W at a junction temperature of about 95 °C.

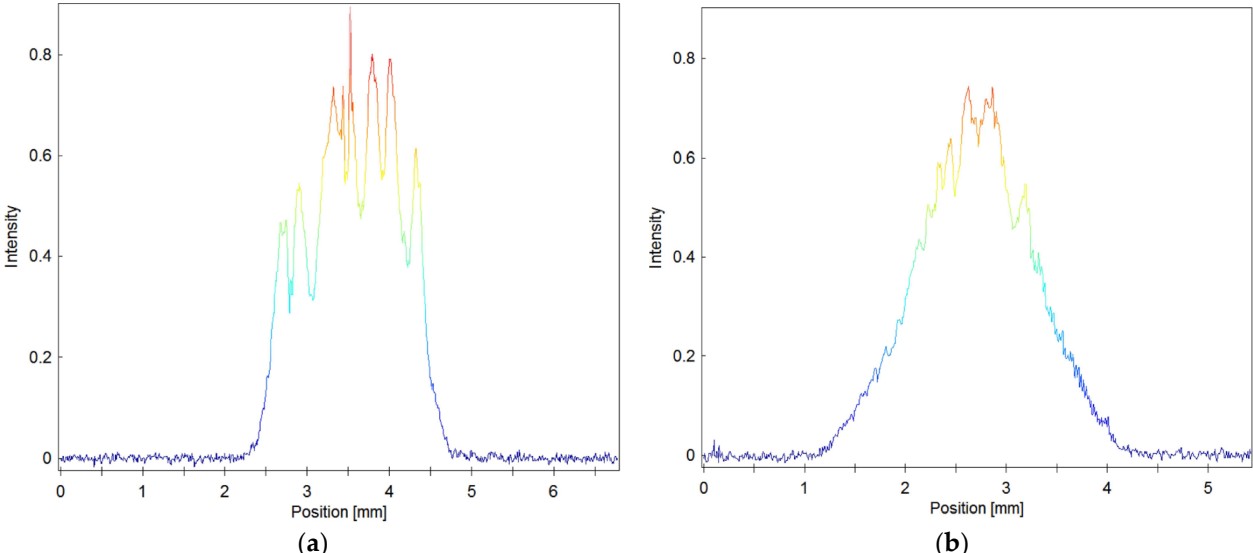

**Figure 2.** Blue LD beam profile characteristics: (**a**) X cross section; (**b**) Y cross section.

Figure 3 shows the results of the blue laser diode light output and junction temperature characteristics over time, and the junction temperature is calculated according to the JEDEC 51-1 standard. Here, the blue laser diode k-factor is 0.0011948 V/°C. It was confirmed that the light output of the blue laser diode decreased as the junction temperature increased. When the junction temperature was 25 °C, the light output of 4.423 W was confirmed at about 95.79 °C after 60 min. In addition, the light output was 3.822 W, which confirmed the characteristic of reducing light output by about 13.59% compared with the initial temperature of 25 °C. In consideration of this light output reduction characteristic, when manufacturing a blue laser white light module, a heat dissipation structure design that can smoothly dissipate heat generated from a blue laser diode and a phosphor is required.

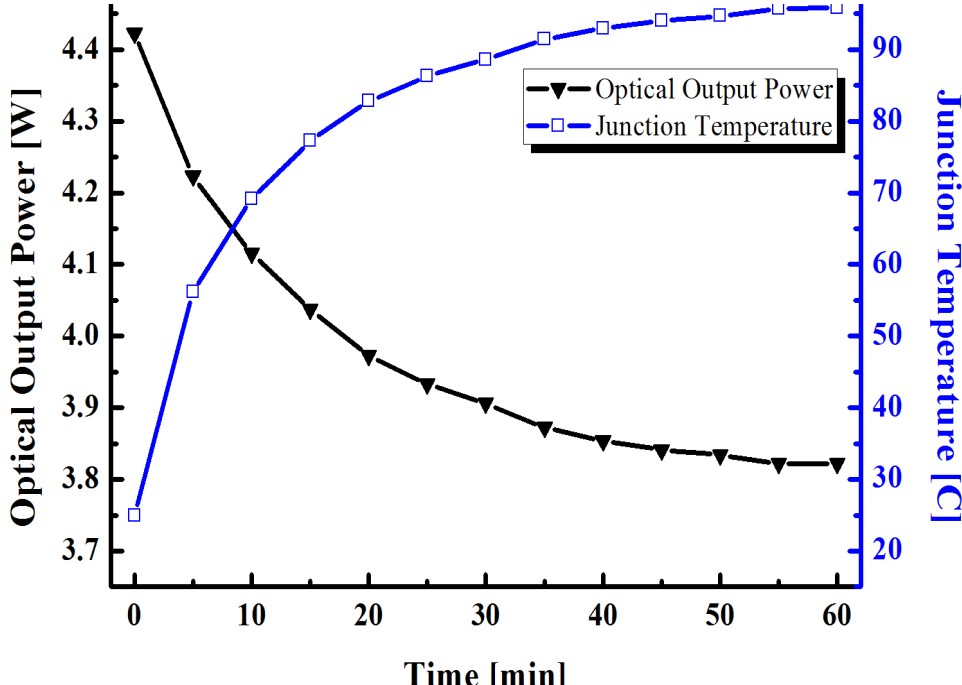

**Figure 3.** Blue laser diode light output and junction temperature characteristics over time.

### 2.1.2. Lateral Displacement Part Design

In order to emit white light by irradiating blue LD beams arranged in parallel to remote phosphor, it is necessary to reduce the beam spacing of LDs diverged at 11 mm intervals. A plurality of mirrors are used to keep the beam spacing parallel and reduce the beam spacing. However, this method has a problem in that the structure of the optical system becomes complicated because of the increase in the number and arrangement of the optical system. In order to improve this structure, an optical design was conducted to reduce the beam spacing using the total internal reflection characteristics of the rhombus prism, as shown in Figure 4.

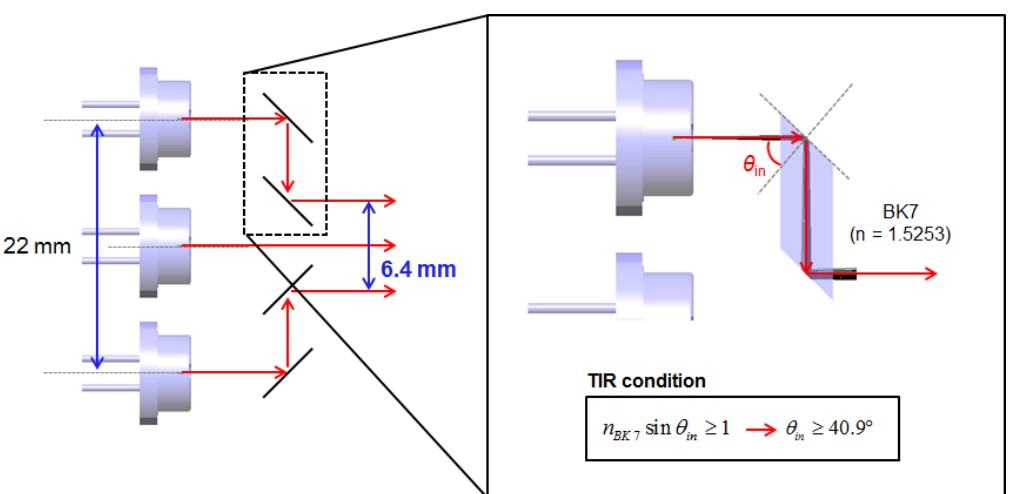

**Figure 4.** Simplified optical system structure using the total reflection characteristics of a rhombus prism.

There is a limit to implementing the beam shape of the divergence angle of the blue laser diode in an accurate nonlinear Gaussian shape like the beam profile measurement result. Thus, in order to overcome the limitations of the implementation of the Gaussian shape, it was set as the surface source. As a result, the characteristics of two types of divergence, $\theta_{full} = 2°$ and $\theta_{full} = 4°$, were confirmed.

Figure 5 shows the beam size change through a rhombus prism lens. As a result of applying a rhomboidal prism lens instead of the existing pair of mirror structures, the optical axis could be moved and the beam spacing could be moved from the existing 22 mm to 6.4 mm. In addition, it is possible to implement simplified mirror optical components, which are judged to contribute to improving the reliability of optical components.

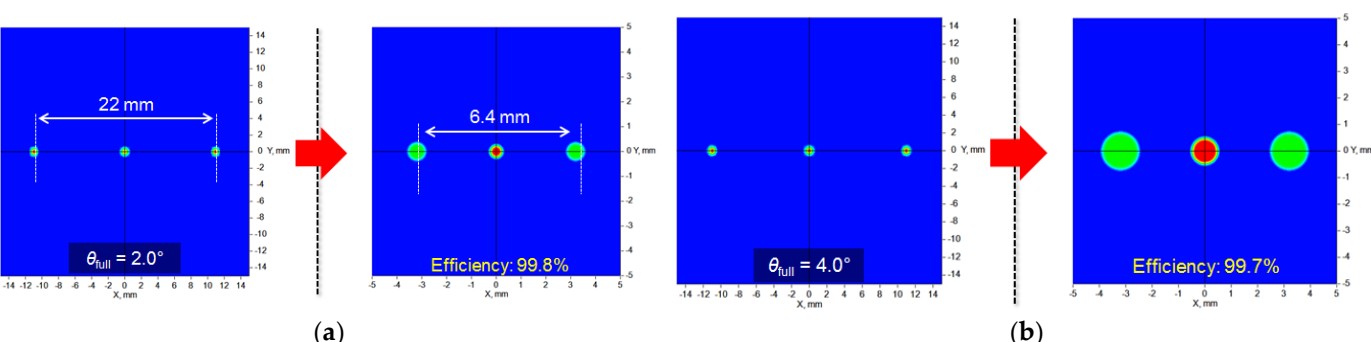

**Figure 5.** Beam spacing and size characteristics according to the divergence angle of a rhombus prism: (a) $\theta_{full} = 2°$; (b) $\theta_{full} = 4°$.

It was confirmed that the size of the laser beam propagated through the rhombus prism increases in proportion to the distance and beam angle. In addition, as the divergence angle of the beam increases from 2° to 4°, the efficiency characteristics were confirmed to decrease by 0.1% from 99.8% to 99.7%. At this time, the size of the linewidth was set to 0.8 mm in diameter.

### 2.1.3. Fold Mirror Part Design

In order for the blue laser beam to be irradiated to the remote phosphor, the beam path must be changed through the fold mirror and the white light excited from the phosphor must be transmitted through the fold mirror. A dichroic filter was applied to satisfy these bi-directional optical conditions. The range of the reflection wavelength of the dichroic filter is 350 to 470 nm, and the reflectance characteristic at this time is 98% on average. The range of transmission wavelength was 490–850 nm and the transmittance characteristics were defined as 90%. Figure 5 shows the simulation results by applying the dichroic filter. The blue laser beam propagated through the rhombus prism is reflected by a dichroic filter having an inclination angle of 45°, the beam path is changed and propagated toward the phosphor, and a portion passes through the dichroic filter. In Figure 5, the size of the pre-optics was confirmed based on the distance of 22~23 mm between the center of the dichroic filter and the remote phosphor.

In order to implement the overall structure in the form of a square, $f_{EFL}$ of at least 10 [mm] or more ($R \geq 5$) must be satisfied, and the relationship to the focal length is as shown in Equation (1).

$$f_{EFL} = \frac{1}{f} = (n_{bk7} - 1) \times \frac{1}{R} \tag{1}$$

In this case, Lambertian beam shaping optimization of the phosphor for the planoconvex lens should be performed. In Figure 6, a plano-convex lens with $f_{EFL}$ of 20 [mm] is used to focus the blue laser beam and structure to irradiate it to the phosphor.

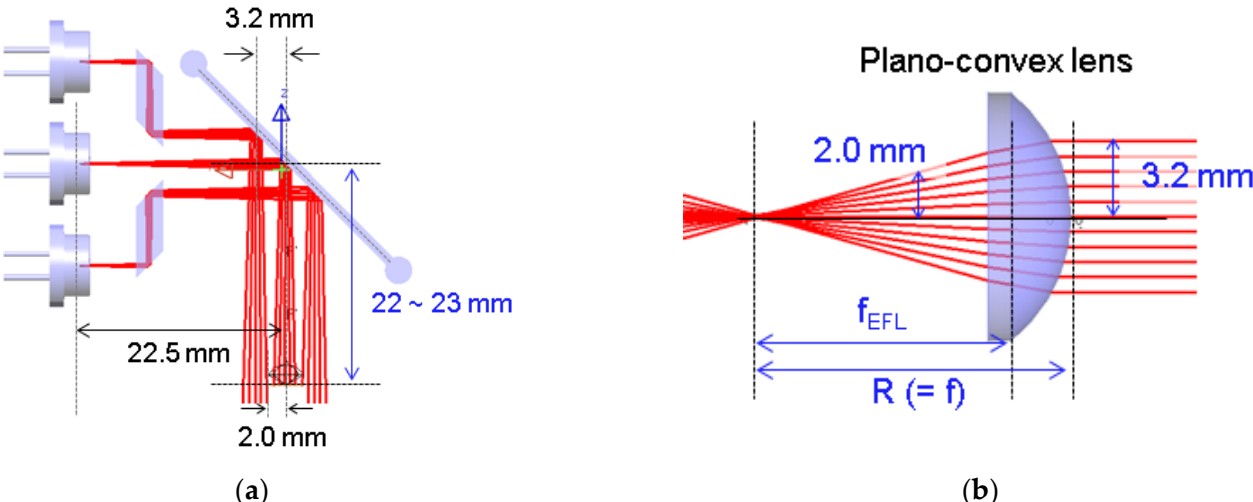

(**a**)    (**b**)

**Figure 6.** Beam path and phosphor irradiation focal length characteristics with dichroic filter applied: (**a**) changes in blue laser beam path; (**b**) characteristics of initial detection distance according to beam width.

### 2.1.4. Pre-Optics Part

Pre-optics refers to an optical structure that condenses the beam of a blue laser diode, irradiates it to a phosphor, and injects white light excited by the phosphor into a dichroic filter size.

Figure 7 confirms that $f_{EFL}$ irradiates 4 mm × 4 mm remote phosphor from blue LD using a planar convex lens with 20 mm characteristics.

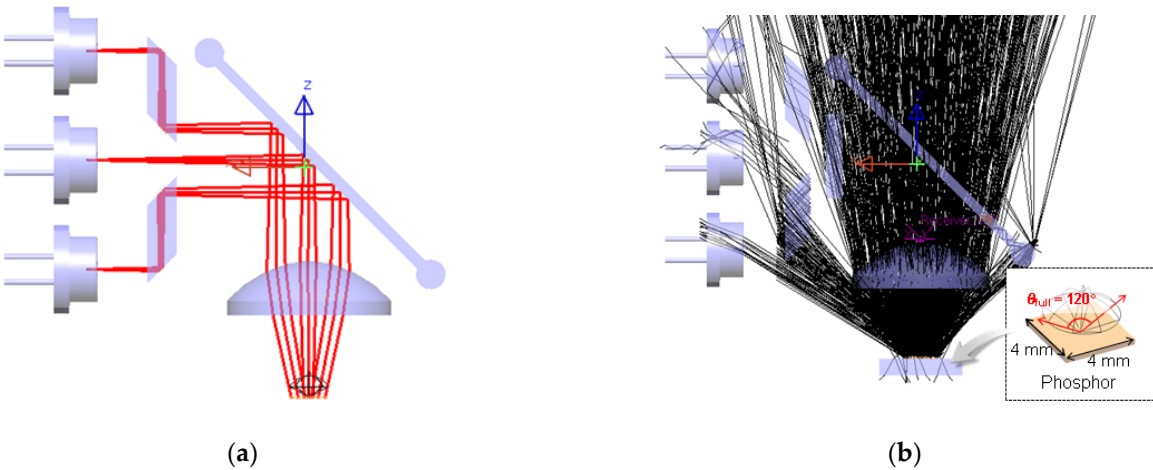

(**a**)　　　　　　　　　　　　　　　　　　　　　　　　　　(**b**)

**Figure 7.** Phosphor and dichroic filter incident beam characteristics using a planar convex lens: (**a**) incident path from the laser diode to the phosphor; (**b**) incident ray characteristics to the dichroic filter.

However, it was also confirmed that one planar convex lens could not accommodate the light-emitting area pattern emitting from the phosphor to Lambertian.

In this paper, a light-emitting surface (LES) optical system structure was proposed to solve the problem of out-of-light receiving range. The size of the LES was limited to 7.2 mm in diameter and it was designed to minimize the reduction in efficiency reflected from the inside by applying a reflective coating on the inside. Figure 8 shows an optical structure with limited beam size, where the LES size was limited to a diameter of 7.2 mm.

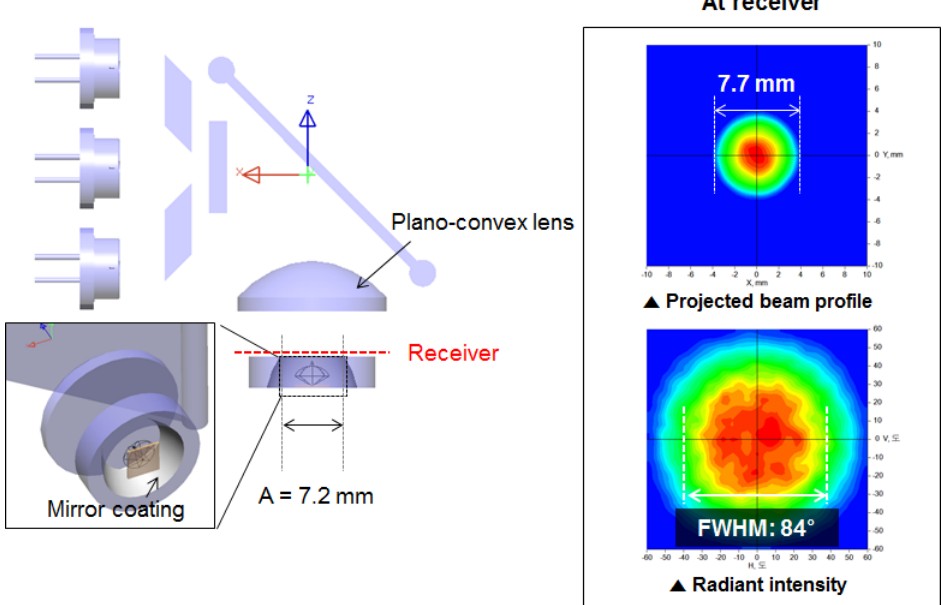

**Figure 8.** Proposed optics limited to a beam size of 7.2 mm.

Figure 9 shows the simulation results with the beam size limited to 7.2 mm. It was confirmed that the white light emitted from the phosphor was received on the lens incident surface in the plano-convex lens with a diameter of 16.5 mm. In addition, a result that could not be accommodated within the dichroic filter size was confirmed owing to the characteristics of the light distribution angle of 40° or more.

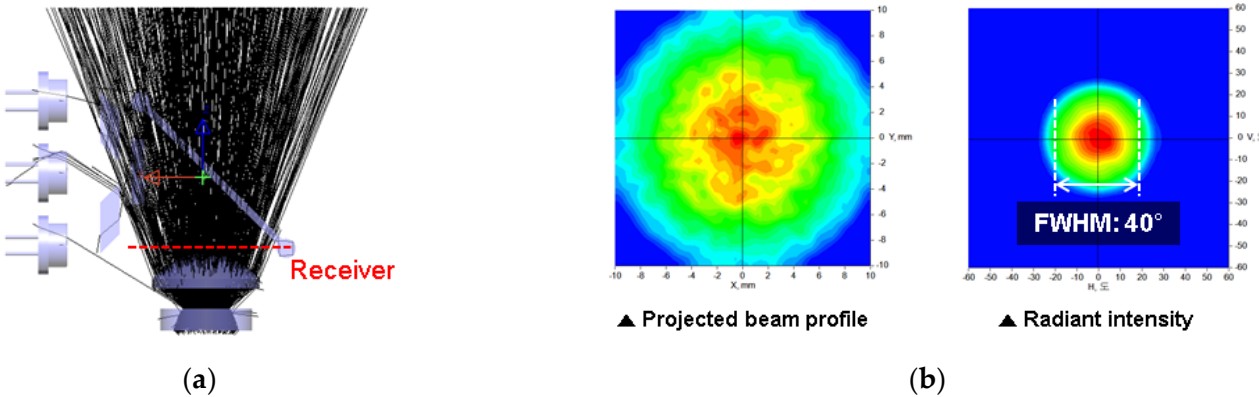

(**a**)
(**b**)

**Figure 9.** Simulation characteristics with limited LES optics: (**a**) ray path characteristics; (**b**) beam profile and light distribution characteristics.

In order to solve the characteristic of the light distribution angle of more than 40°, a planoconvex lens with a focal length of 16 mm $f_{EFL}$ was additionally inserted between the limited optical parts and the planoconvex lens. Figure 10 shows the optical structure of two planar convex lenses. As a result of the simulation, the size of the beam profile decreased from 20 mm in diameter to 16 mm before application, and there was confirmation that the light distribution angle characteristics improved from 40° to 23°.

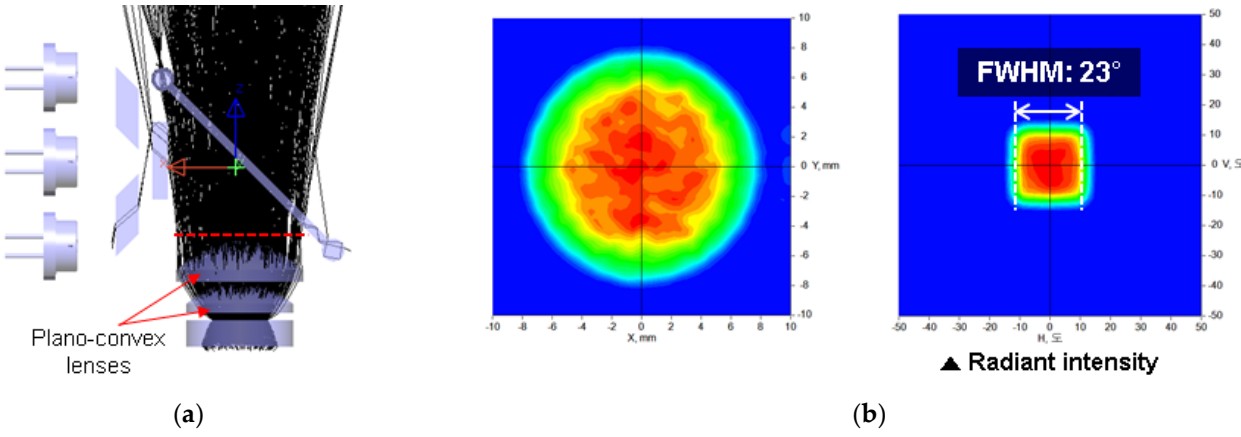

(**a**)
(**b**)

**Figure 10.** Simulation characteristics using a two-stage plano-convex lens: (**a**) improved ray path characteristics; (**b**) improved beam profile and light distribution characteristics.

Figure 11 shows the final designed bidirectional blue laser white light source module by applying a dichroic filter. Figure 11a shows the path through which the white light excited by the phosphor exits through the final planar concave optical system. In Figure 11b, the blue laser beam transmitted through the dichroic filter is irradiated to the limited diffusion component, then the diffused blue laser beam is emitted by the pre-optics. It shows the result of being reflected by the dichroic filter, mixed with the white light, and emitted through the plano-concave lens.

Figure 12 shows the beam path of the laser module manufactured based on the light source module design and experimental results. It shows the path of the blue laser beam equivalent to the simulation result.

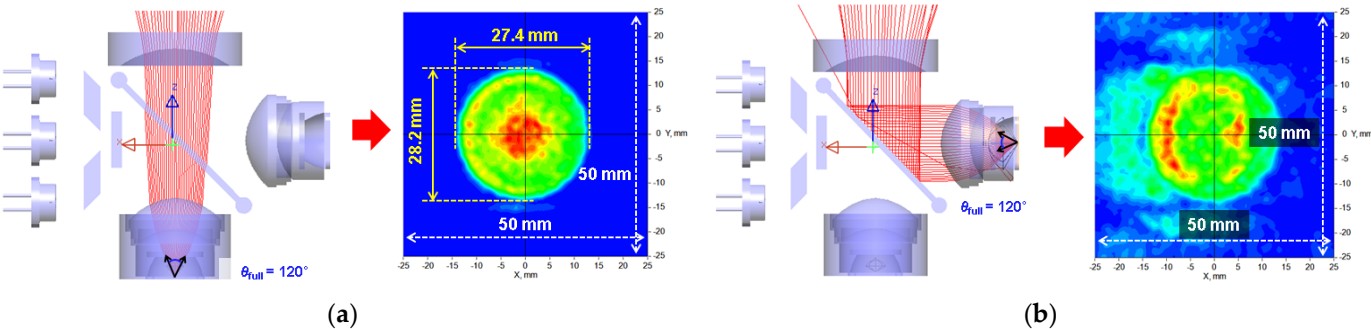

(**a**)  (**b**)

**Figure 11.** Bidirectional blue laser white light source module with the final designed dichroic filter applied: (**a**) phosphor white light path; (**b**) transmitted blue laser beam path.

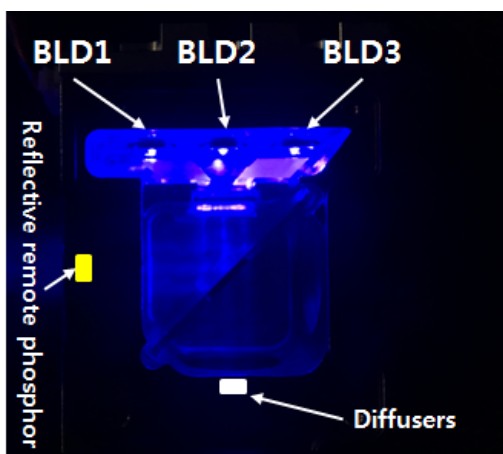

**Figure 12.** Proposed white light module optical design.

Beams diverged from blue laser diodes 1 and 3 are incident to the rhombus prism optical system and irradiated to the dichroic filter by changing the beam path and spacing, and blue laser diode 2 is directly irradiated to the dichroic filter. It was confirmed that most of the blue laser diode beam irradiated to the dichroic filter was reflected and the light path was changed in the 270° direction to be irradiated to the spaced phosphor, and some of the beam irradiated from the dichroic filter was refracted by the thickness of the filter. After that, the characteristics of being transmitted and irradiated in the direction of the diffusion component were confirmed.

### 2.2. Searchlight Collimated Beam Optical Design

A beam expander is an optical system that expands a collimated input beam into a collimated output beam with a larger diameter, and is a focusing optical system developed based on the concept of an optical telescope. In this optical system, light rays of an object are incident and emitted parallel to the optical axis of the internal optical system.

In the beam expander, the objective lens and the image lens are positioned opposite to each other. A Kepler-style beam expander must be designed so that the collimated input beam is focused to a point between the objective lens and the image lens.

As a result, the energy of the laser is focused to create a point within the system. In order to improve the divergence angle through the laser beam expander, magnifying power must be improved.

The divergence angle of the beam magnifier was confirmed to be improved through the use of a large-area collimating lens.

A lens with a large area more than 10 times that of the light source was reviewed and a system was constructed by combining aspheric condenser lenses and achromatic lenses.

Figure 13 shows the structure of the large-area collimating optical system, and the performance evaluation for the size and incident angle of the light source was verified through simulation. The size of the light source was 5 × 5 [mm] and 7 × 7 [mm], and the incident angles were verified at 40°, 60°, and 80°.

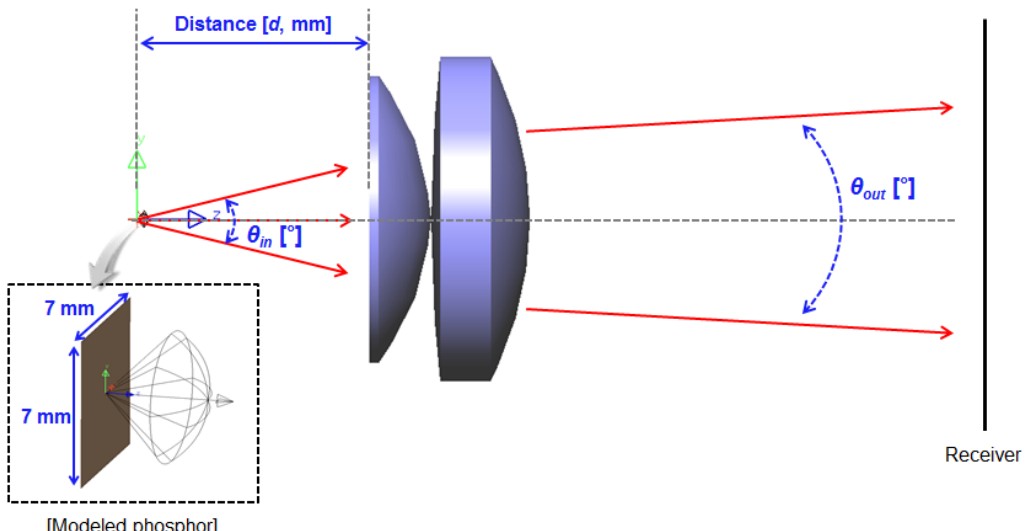

**Figure 13.** Large-area collimating optical system structure.

Table 2 shows the performance characteristics of the large-area collimating optical system. It was confirmed that the characteristics of 5 × 5 [mm] and 7 × 7 [mm] did not change the emission angle as the incident angle increased. The efficiency was about 80.1 [%] at 40°, 76.9 [%] at 60°, and 46.7 [%] at 80°. In addition, the efficiency of 7 × 7 [mm] was confirmed to be about 80.1 [%] at 40 °, 76.5 [%] at 60 °, and 47.4 [%] at 80 °.

**Table 2.** Performance characteristics for large-area collimating optics.

| Source Size | $\theta_{in} = 40°$ | $\theta_{in} = 60°$ | $\theta_{in} = 80°$ |
|---|---|---|---|
| | At d = 88.6 mm | At d = 84.6 mm | At d = 85.3 mm |
| 5 × 5 [mm] | $\theta_{out}$ = 2.4° | $\theta_{out}$ = 2.4° | $\theta_{out}$ = 2.4° |
| | Efficiency: 80.1% | Efficiency: 76.9% | Efficiency: 46.7% |
| | At d = 88.5 mm | At d = 84.8 mm | At d = 83.8 mm |
| 7 × 7 [mm] | $\theta_{out}$ = 3.3° | $\theta_{out}$ = 3.3° | $\theta_{out}$ = 3.3° |
| | Efficiency: 80.1% | Efficiency: 76.5% | Efficiency: 47.4% |

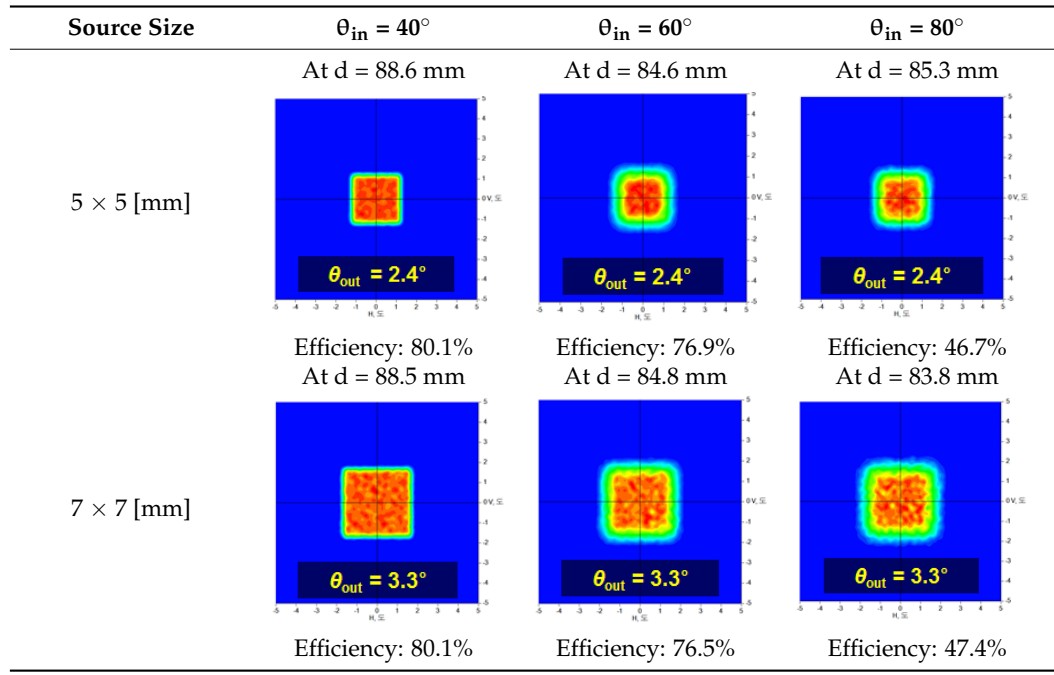

As for the characteristic that the efficiency characteristic decreases as the incident angle increases, even if an optical lens 10 times larger than the light source is applied, the divergence angle versus the distance is not significantly improved. This is because all but the accommodating area of the secondary optical system occurs as a loss owing to the characteristics of the clear aperture of the secondary optical system and the characteristics beyond the lens diameter.

For the ideal parallel beam control for long-distance propagation, the key design technology is to minimize the size of the light source while narrowing the divergence angle diverging from the light source.

As a result of reviewing the performance characteristics of the large-area collimating optical system, it was confirmed that a divergence angle of less than 4° can be secured even for incident light with a wide divergence angle when using a lens with magnification.

The divergence angle was reviewed by applying a large-area collimating lens to the dichroic filter-based bi-directional optical system structure.

Figure 14 shows the laser white light module large-area collimating optical system structure, and the divergence angle characteristics according to the distance L were confirmed.

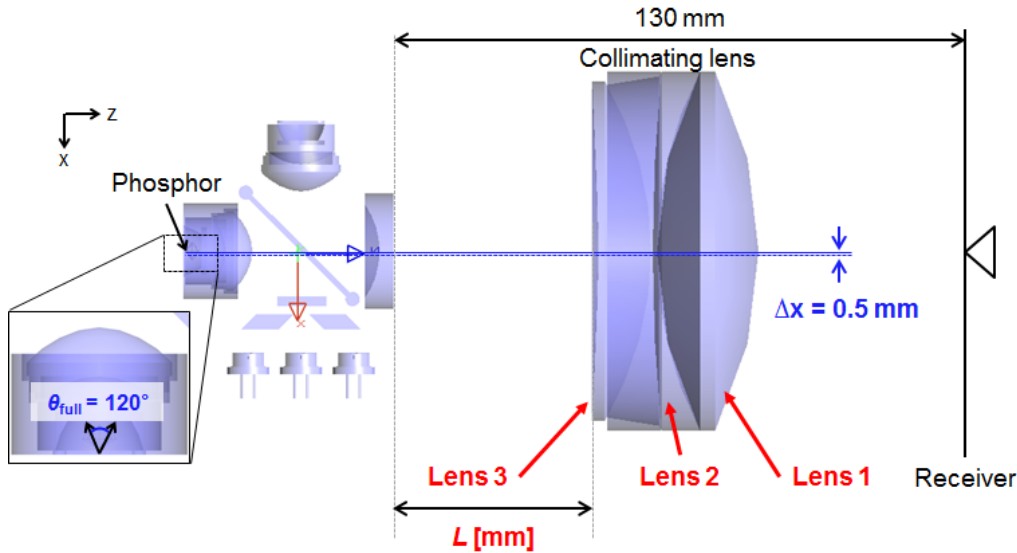

**Figure 14.** Laser white light module large-area collimating optical system structure.

The large-area collimating lens consists of three lenses. Lens 1 and Lens 2 are composed of achromatic lenses with a diameter of 75.0 [mm], and Lens 3 are composed of positive meniscus lenses with a diameter of 70.1 [mm]. The distance between the achromatic lens and the positive meniscus lens is 26.25 [mm].

Figure 15 shows the characteristics of the divergence angle of the laser module optical system for each distance. At 25 [mm], it shows the characteristic of about 12°; as the distance from the light source increases, the divergence angle narrows. Within the 42.5 [mm] section, a divergence angle of 4 degrees or less was satisfied. In addition, although the size of the incident light source increases relatively, the application of a large-area collimating lens satisfies the performance without a significant change in the incident angle.

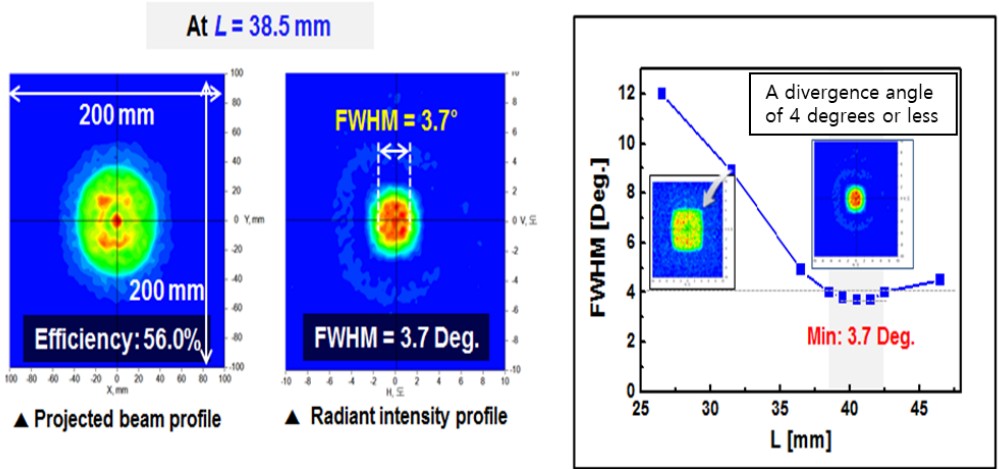

**Figure 15.** Laser module optical system divergence characteristics by distance.

### 3. Conclusions

In this paper, an optical structure using a high-power blue LD and remote phosphor was proposed and a white light module was designed.

In addition, a parallel beam for a searchlight was designed by applying the proposed white light module and a bidirectional large-area collimating lens based on a dichroic filter.

The proposed white light module applied high-output blue LD, and optimally designed the lateral displacement part, fold mirror part, and pre-optics part. The design of the lateral displacement part improved the large width from 22 mm to 6.4 mm by moving the optical axis by applying the total reflection characteristics of the rhombus prism. The design of the fold mirror part applied a dichroic filter so that the feel was 20 [mm]. The technology applied here uses the blue laser beam with a flat convex lens so that the phosphor is irradiated. In the design of the pre-optics part, a dichroic filter was applied to obtain a bidirectional blue laser white light source. In order to use the designed white light module as a searchlight, it is necessary to design a narrow divergence angle. In this paper, a collimated beam for a searchlight was designed using a narrow divergence angle by applying a bidirectional large-area collimating lens based on a dichroic filter. As a result, it was possible to secure a low divergence angle of 4 degrees or less.

Future research will proceed with the design of heat dissipation components to minimize the change in the response characteristics of the light efficiency decrease owing to the high temperature of the blue laser beam.

**Author Contributions:** Lighting filer design, S.-M.L.; software, J.-H.C.; LightTools simulation, W.-B.L. All authors have read and agreed to the published version of the manuscript.

**Funding:** This paper was supported by Wonkwang University in 2020.

**Conflicts of Interest:** The authors declare no conflict of interest.

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
