# Peer review of "A Study on the Development of a White Light Source Module for a Large-Capacity Searchlight Using a Blue Laser Diode"

_electronics, doi:10.3390/electronics12030760_

Round 1

Reviewer 1 Report

The authors of the proposed article tried to design their own variant of a high-power searchlight based on luminophore exciting by means of  3 high-power blue laser diodes laser. The authors demonstrated a huge amount of work, but in my opinion, the proposed article has not enough novelty. All used in design components and principles are well known. Moreover, the article was written in a not enough clear and very hard to reading manner.  For example - the second sentence in the abstract (lines 12,13,14) sounds: "The proposed white light module optical structure is a bi-directional optical system structure based on a rhombus prism lens to satisfy the structure of a reflective phosphor" - the word "structure" repeats 3 times! Then, in line 15 of the abstract one can meet the term "movement path of laser beam"- the authors mean laser beam propagation path I guess. What does it mean "FL" and "range of 38.5 to 42.5 [mm]" in 22-23 lines of the abstract? In line 11 of the abstract authors mentioned the thermal stability of the designed module, but only in line 287 of the conclusions promised to consider this item in their future investigations. Further, in the abstract line 14 and line 115 in the text authors mentioned rhombus prism lens and rhomboidal prism lens respectively. In line 109 this element is called just the rhombus prism and in line 277 it transforms into a rectangular prism.

Line 201-202: "It was confirmed that the path of the blue laser beam reflected through the dichroic filter was moved to the right and irradiated". Does it mean that the path was moved to the right and irradiated? :) . Line 135: "light movement path is changed vertically..." light does not move - it propagates! Vertically with regard to what?

So on and so on.

In the introduction, the authors did not mention the possible alternative laser diode white light sources so-called RGB laser sources which are based on the direct mixing of light from red, green, and blue light from high-power laser diodes. Such RGB lasers are widely presented in the market.

Authors widely used computer simulation in their investigation but did not indicate the software name. The results of the simulation presented in the figures (for example fig. 6. b) are not enough clear- the density of beam tracks is too high.

Fig. 11 is also not clear, it needs additional explanations.

Fig.14 includes non-Latin characters.

So I propose rejecting the article from publication.

Author Response

The completion of our thesis seems to have improved due to the faithful review of the judges. The points pointed out by the judges were revised and supplemented as follows and reflected 100% in the original paper. Thank you.

Responses to the review opinions will be submitted as attachments.

Reviewer 2 Report

"A study on the development of a white light source module for 2 a large-capacity searchlight using a blue laser diode" by Lee et al. submitted in Electronics.

The manuscript presents an optical structure using a high-power blue laser diode and remote phosphor, and also, a white light module. The manuscript needs some revisions as follows:

1. All symbols and abbreviations should be described, for example, FL in the abstract.

2. The manuscript needs to be polished. There are some typos and grammatical errors.

3. Please give more details for Fig. 1 (subsection 2.1 should be completed).

4. Is there any information on the linewidth of optical sources? please clarify.

5. The experimental temperature of 25° C is considered. The temperature dependence effect on the output should be considered and addressed.

6. Please give a brief comparison between the results of this manuscript and previous works.

7. The references are a bit old and more recent works should be addressed.

8. Finally, it seems that the literature survey is not complete, and the introduction needs to be completed.

Author Response

(The authors gave the same response as above.)

Reviewer 3 Report

1-      The authors should explain what makes this work unique and how it differs from previous research.

2-      Please discuss in the text the reproducibility of the data in Figs. 2 and 4.

3-      Please describe in the text how you monitor the beam profile in Fig.2.

4-      Please describe in the text how you monitor the beam size changes in Fig. 4.

5-      Please describe in the text how you perform simulation results in Fig. 8.

6-      In the text, please explain how the rhombus prism might affect the beam profile and the LD output power.

7-      It is necessary to add clarification and in-depth discussion to the text for the following issues:

·         The efficiency characteristics in Figure 4.

·         The factors that could affect the stability of the proposed white light module

Also here are few typographical errors:

·         Page 1, line 15 – “band-pass filter of 500 [nm]” should be replaced by " band-pass filter of 500 nm".

·         Page 1, line 23 – “of 38.5 to 42.5 [mm]” should be replaced by " of 38.5 to 42.5 mm".

·         Page 1, line 38 – “and reliability.[1-3].” should be replaced by " and reliability [1-3].".

·         Page 6, line 178 - " 16mm fEFL" should be replaced by " 16mm fEFL".

Author Response

(The authors gave the same response as above.)

Reviewer 4 Report

The introduction section should be completed.

For proposed construction of white light source the technology of phosphor plate coupled with mirror seems to be critical. There are  few information in publications of similar construction. But some exists and the technology of phosphor-mirror technology should be mentioned and compared at least with positions:

M. Borecki, P. Doroz, P. Prus, P. Pszczolkowski, J. Szmidt, M.L. Korwin-Pawlowski, J. Frydrych, A. Kociubinski, M. Duk, „Fiber optic capillary sensor with smart optrode for rapid testing of the quality of diesel and biodiesel fuel”, International Journal On Advances in Systems and Measurements vol 7, pp. 57-67, 2014. – Phosphor type and approximated height should be compared and pointed, as well as used wavelengths.

Mu-Huai Fang, Zhen Bao, Wen-Tse Huang, and Ru-Shi Liu, “Evolutionary Generation of Phosphor Materials and Their Progress in Future Applications for Light-Emitting Diodes”  Chem. Rev., 122, pp. 11474–11513, 2022 -- Phosphor type and its parameters should be compared.

Lin, YC., Karlsson, M. & Bettinelli, M. “Inorganic Phosphor Materials for Lighting”. Top Curr Chem (Z) 374, 21 (2016). https://doi.org/10.1007/s41061-016-0023-5 --- WLEDs based on a a NUV-LED and a mixture of RGB phosphors should be mentioned and compared to a blue LED combined with a yellow phosphor.

Minor corrections:

Figure 1:

The dichroic mirror parameters should be pointed. Optical paths on selected wavelengths (blue and with) should be visualized. Optoelectronics components have to be labeled. The output of white light source have to be pointed.

Figure 2:

Crossection X of laser beam looks odd. Can it be  explained ?

Author Response

(The authors gave the same response as above.)

Round 2

Reviewer 1 Report

After introducing corrections the article looks much better. But I still doubt the novelty of the presented results. As I have mentioned in a previous review all used in design components, principles, and technical decisions are well known. But maybe the synthesis of all these known things to the developed device could be considered as some novelty - it is up to the editor's decision.

Author Response

The judges have no additional comments. However, thank you for reviewing the revised paper again.

Reviewer 2 Report

The responses to the following comments are not clear and adequate:

Comment 4. Is there any information on the linewidth of optical sources? please clarify.

Comment 5. The experimental temperature of 25° C is considered. The temperature dependence effect on the output should be considered and addressed.

Comment 7. The references are a bit old and more recent works should be addressed.

Comment 8. Finally, it seems that the literature survey is not complete, and the introduction needs to be completed.

The authors have mentioned "The revised manuscript reflected the revisions of the reviewers". Where? Why? How? ...

Author Response

The completion of our thesis seems to have improved due to the faithful review of the judges. The points pointed out by the judges were revised and supplemented as follows and reflected 100% in the original paper. Thank you.

The responses to the following comments are not clear and adequate:

Comment 4. Is there any information on the linewidth of optical sources? please clarify.

Answer : It was reflected in the revised paper line [176-177]

Comment 5. The experimental temperature of 25° C is considered. The temperature dependence effect on the output should be considered and addressed.

Answer : It was reflected in the revised paper line [131-144]

Comment 7. The references are a bit old and more recent works should be addressed.

Answer : It was reflected in the revised paper line [381-387] and line [396-399]

Comment 8. Finally, it seems that the literature survey is not complete, and the introduction needs to be completed.

Answer : It was reflected in the revised paper line [54-57] and line [83-85]

Reviewer 3 Report

In response to my previous suggestions and concerns, the authors have made reasonable changes to the manuscript. Overall, the manuscript reads well and clarifies the authors' work. In my opinion, the manuscript contains currently all information and is ready for publishing in the Journal "electronics" as a regular article.

Author Response

(The authors gave the same response as above.)

Round 3

Reviewer 2 Report

In response to comment 4, the size of the diverging surface source is mentioned instead of the "linewidth", in lines 176-177. However, the manuscript can be accepted for publication.

Author Response

The second review by the reviewers seems to have further improved the completion of the thesis. The points pointed out by the judges were corrected and supplemented as follows and 100% reflected in the original text. Thank you.

In response to comment 4, the size of the diverging surface source is mentioned instead of the "linewidth", in lines 176-177. However, the manuscript can be accepted for publication.

 Answer : It was reflected in the revised paper line [176-177]
